# Optimal Politics of Conflict over Physical-Industrial Development Using a Technique of Cooperative Game Theory in Iran

**Samaneh Zahedi [1], Amir Hedayati Aghmashhadi [2],\* and Christine Fürst [3]**

[1] Environmental Management, Islamic Azad University, Tehran Science and Research Branch, Tehran 1477893855, Iran; smnhzahedi@yahoo.com

[2] Department of Environmental Science and Engineering, Faculty of Agriculture and Environment, Arak University, Arak 3848177584, Iran

[3] Department of Sustainable Landscape Development, Institute of Geosciences and Geography, Martin Luther University Halle-Wittenberg, German Centre for Integrative Biodiversity Research (iDiv) Halle-Jena-Leipzig, 06120 Halle, Germany; christine.fuerst@geo.uni-halle.de

\* Correspondence: a-hedayati@araku.ac.ir; Tel.: +98-9112569731

**Abstract:** One of the most important parameters for economic growth is industrial development in many developing regions like Iran. The Markazi province in the center of Iran is one of the most important industrial areas in the country, where unplanned economic development in recent decades has led to many social and environmental problems. Accordingly, the main organizations involved in industrial development in this region are facing difficulties in the future development of industrial areas, which has become a complex problem. Therefore, the main purpose of this paper is to study the industrial development dispute in the Markazi province through a cooperative model of game theory in order to achieve conflict resolution through a comprehensive approach. In this research, the conflict has been analyzed through strategic analysis of stakeholders. For this purpose, a model of cooperative game theory and its bargaining analysis methods, including social choice rules (SCRs) and fallback bargaining (FB), and six available options were used. According to the six SCRs, the most likely option that can exist between the Department of the Environment (DOE) and the Industrial and Mining Organization (IMO) is compromise coordination (C). In addition, the results of the Fallback Bargaining (FB) rule in three different forms show that the most appropriate options for agreement between the IMO and DOE are the moderate version of construction through arbitration (TCa) and compromise coordination (C). In fact, the results indicate that if the actors involved in this conflict do not cooperate to resolve it, it can lead to more complex problems and the involvement of other groups who may not even have a proper perception of the conflict. In addition, the findings show that cooperation between the parties and understanding of their positions and views along with the policy coherence are necessary to strive for sustainable development and maintain economic growth and development.

**Keywords:** industrial development; conflict resolution; cooperative game theory; Iran

## 1. Introduction

The emergence and proliferation of advanced digital manufacturing (ADP) technologies such as artificial intelligence, big data analytics, cloud computing, the Internet of Things (IoT), advanced robotics and additive manufacturing have radically changed the nature of manufacturing production [1]. Under the right conditions, the adoption of these technologies by developing countries can promote inclusive and sustainable industrial development (ISID) and the achievement of the Sustainable Development Goals (SDGs) [1].

Land is an important prerequisite for physical industrial development [2]. This finite resource is central to human welfare and livelihood. However, rapid urbanization in the last century has led to scarcity of land and escalation of land prices [2].

It has also led to widespread land use changes due to the expansion of cities and industrial areas [3]. Urban, rural and industrial developments can have profound impacts on the surrounding environment. Such impacts can defeat the purpose of development so that the disadvantages can outweigh the benefits [3].

Industrialization has the potential to contribute to the achievement of a number of social goals, such as employment, poverty reduction, gender equality, labor standards and better access to education and health care [4]. At the same time, however, industrial processes can have negative impacts on the environment, causing climate change, loss of natural resources, air and water pollution, and species extinction [3]. These factors threaten both the global environment and economic and social well-being [4]. The emergence of many new industrial products poses a major regulatory and control challenge. Their emissions are often neither regulated nor qualified, leading to unknown environmental and health impacts. For example, the emissions industry has seen an increase in pollutant emissions globally between 1990 and 2014 [5].

Decision-making processes, particularly in relation to land and property development, are complex processes [6–8]. First, different groups of actors or stakeholders are involved in the process (e.g., land owners, developers, different government agencies, investors, end users and NGOs). Second, they respond to each other's strategies differently and with different nationalities [9]. In this paper, this complexity is taken as the starting point for analyzing decision making in the development of industrial land in Iran's Markazi province (Figure 1). The interactions between actors in the development of industrial land can be studied through a case study analysis. However, one of the drawbacks of this approach is the limited number of successful cases. Moreover, case study analysis might pose some difficulties regarding the influence of individual factors on the results. This would lead to difficulties in generalizing the findings on stakeholder behavior.

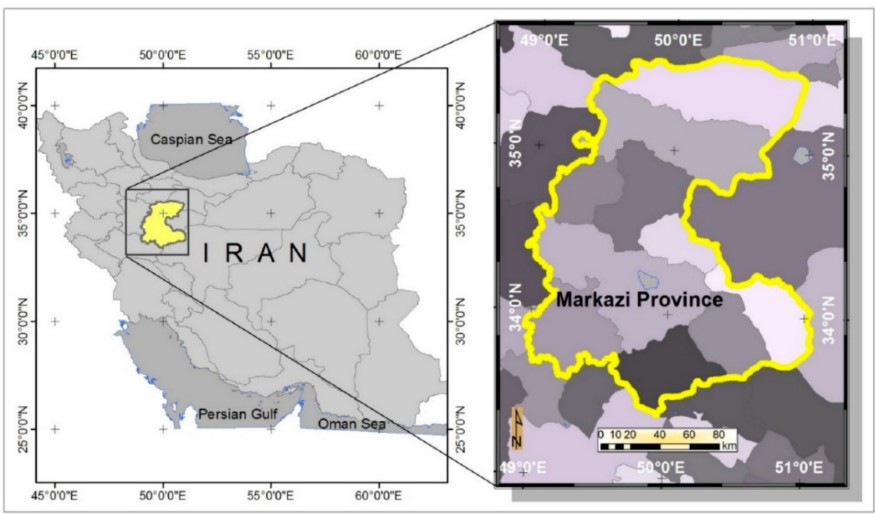

**Figure 1.** Location of the Markazi province in Iran.

### 1.1. Why Game Theory?

Game theory is a theory of interdependent decision making in which the decision makers involved have opposing preferences and the outcome of their decisions cannot be determined by only one party or actor [9]. The term "game theory" stems from the similarity of collective decision-making situations to well-known parlor games such as chess, poker, and Monopoly [10]. Although game theory is rooted in decision theory [10], there is a clear difference between the two. In decision theory, decision processes are usually analyzed from the perspective of a single actor, whereas game theory focuses its analysis on the interaction between many actors [10]. Because game theory focuses on situations in which interactions and interdependence play a role, it can be considered an extension of decision theory [9].



Because of its focus on conflicting preferences, game theory is often defined as conflict theory [11,12]. Aumann (1989) has even suggested referring to "interaction decision theory" instead of game theory because the former more accurately describes the content and focus of the theory [11]. Modern game theory began in the 1920s with the work of [13,14]. However, it was after the seminal work of [15] that game theory sparked a wave of interest, especially among mathematicians and economists. Since then, game theory has had a lasting impact on other areas of the natural sciences, such as biology, physics, and computer science, as well as the social sciences, including anthropology, psychology, sociology, politics, and philosophy [16–18]. With regard to land development processes, the number of applications of game theory has been limited [19,20]. The lack of attention can perhaps be explained by the fact that land and property development and spatial planning are highly contextual. The issues of pluriformity, complexity and interdependence have gained much prominence in urban societies and hence in the current planning literature [21–23]. Modeling these processes sometimes makes no sense because the interactions of the actors involved are undeniably too complex and so dependent on case-specific circumstances. It may therefore seem strange to argue for modeling industrial land development processes in simplified models. However, we make the case nonetheless. This is supported by a review of analytical tools for studying land and real estate development processes by Ball et al. who considers game theory to be a promising method in this area of research [24].

Game-theoretic modeling, like any modeling, implies a simplification and abstraction from the real world. However, in some disciplines, for example in economics, the translation of the real world into a formal model is very much accepted and appreciated. Mathematical approaches to the analysis of complex situations have become a very significant area of study over time [9]. In view of the limitations in modeling complex situations, there are some good reasons to consider the possibilities of game theory as a decision support tool for land development analysis. First, one of the advantages of game theory is that the construction of a model of complex land and real estate development requires a very precise definition of the assumptions about the complex reality underlying the model. In this respect, game theory as a formal model allows one to see exactly how the conclusions of a model follow its assumptions. Of course, the assumptions can and should be debated, but the outcome of game theory modeling can be described very accurately. Furthermore, formal modeling creates a logical structure for an argument and its accumulation. The logical structure of a model allows the modeler to add or remove something to derive new conclusions, allowing for new insights or clarification of an issue. However, without the rigor of a formal model, we might not be able to generalize such an argument to other situations. Moreover, by attempting to create a more general structure of arguments, moving away from complexity can also help to increase the transferability of planning and development strategies at the international level [25].

Second, many scholars have criticized game theory's underlying notion of rational or profit-maximizing players, which assumes complete information for each player. However, recent contributions to game theory seem to offer new opportunities to improve its applicability. In the last decade, game theory has been able to develop more realistic decision models, including models with bounded rationality, models that account for emotions and intuitive decisions, models with incomplete information, and models with asymmetric information positions. In this sense, game theory is leaving its mathematical heritage and increasingly becoming a behavioral theory of decision making, leading to numerous experiments in game theory [26]. The application of game theory to industrial land development and planning can take advantage of this feature. The advantage of this approach is that it provides a simple and clear structure of the situations to be studied. Therefore, in a first stage, it improves our understanding of the decision problems under study. In a later stage, it can be refined and extended to a more sophisticated or realistic model.

Third, the results of previous studies show that it is possible to empirically test and validate the results of game theoretic modeling [27]. Validating the preferences of the actors involved and testing the results of the games to be played can help to increase the

applicability of the results of games in a more general way. A strategic game consists of at least three components: (a) a set of players, (b) a set of actions (for each player), and (c) the preferences for the set of action profiles (for each player) [28].

Newly developed spatial and geographical economies have led to a decentralized approach to industrial policy [29]. Land is one of the most contentious issues and a major source of conflict in developing countries. In most developing and transition countries, many constitutive and regulatory institutions have significant functional deficits [30–32]. Land conflicts can be categorized according to whether they occur at the micro–micro or micro–macro level, i.e., between community groups or between community groups and external governmental, private or civil society organizations [33].

Game theory includes two areas: cooperative game theory and noncooperative game theory. Cooperative game theory analyzes situations in which actors can cooperate to create value by forming coalitions, but also situations in which actors compete to gain value. Their formalism involves a group of actors and a function that specifies the value that each subset of actors (i.e., each coalition) can create on its own (the characteristic function). The characteristic function is the input to a solution concept that returns the value captured by each agent (its imputation) [34]. In cooperative games (also called coalitional games), the central questions are what coalitions will form, and what should be the benefits for the participants. It is usually assumed that the benefits arising from cooperation can be easily distributed and shared among the players. This assumption allows collaboration to be framed as a game with transferable benefits and to apply proven solution concepts from cooperative game theory [35]. A cooperative game is a game with competition between combinations of players (actors), emphasizing that each player should abide by the agreement made by the combinations and obtain the maximum benefit of the combination through mutual cooperation. A cooperative game is about how to create incentives for independent decision makers to act together as a unit to improve their respective status (or utility) in the game [36]. Cooperative game theory is about examining the conditions under which some form of cooperative action might be necessary, and the outcomes that might result. Therefore, there is a clear point of contact between the goals of cooperative game theory and the specifics of land policy [37].

### 1.2. Literature Review

Various approaches have been applied in spatial land-use planning to model the interactions between land-uses and to reach a near-optimal decision. In recent research, optimization algorithms social choice rules (SCRs) and fallback bargaining (FB) models of game theory have been further addressed. In this field, the most commonly used approach is that of employing cooperative algorithms [9,33,37,38]. For example, [9] demonstrates that game theory could help us to identify the key strategic decisions of land and property development projects by showing the different payoffs for stakeholders of their chosen strategies and selecting the equilibrium in which all stakeholders involved are best off; [33] develops a new analytic framework from a behavioral perspective based on game theory. The framework concerns the logic and strategy of conflicts of legal land acquisition, and that of illegal land conversion; [37] seek to explore the underpinning logic of land readjustment using fundamental concepts in cooperative game theory: the Shapley value and the Core. The results shed light on many important practical issues for the policy, ranging from the conditions under which development might be self-initiated to coalition stability, and to the value of an animating agency. In those works, the game theory model was used in the optimization process in order to address some of the computational complexity involved, but such tools are not appropriate to deal with real industrial conflicts because many game theory tools were incorporated into the industrial development goals analysis [29].

The main characteristic of the cooperative theory is the generation of various possibilities. This theory and their methods are usually adopted to optimize land-use layouts in a general way, without incorporating competitiveness. Based on the complexity of industrial development goals conflicts, the most common method for modeling through

the cooperative game theory is social choice rules (SCRs) and fallback bargaining (FB) [39]. In some studies, this tool is also utilized for land-use planning [9,19,23].

### 1.3. The Goal of Study

One of the most important reasons for the sprawl of industrial areas in Iran is the lack of unified planning in the form of national land use planning for industrial areas and the instability of government policies and programs in the industrial sector [38]. This circumstance has led to many problems in the Markazi province, especially due to the lack of consideration of social requirements and the destruction of the environment in recent years.

The purpose of this study is to conduct a comprehensive cooperative examination of the trade-off solutions in the area of policy making among the organizations involved in the industrial development process in the Markazi province with a view of sustainable development because the objectives and management measures of each of the parties involved in this conflict may either lead to worsening environmental and social problems or to reduced economic growth and consequently unemployment in the region. The only solution to the conflict over industrial development in the Markazi province is a collaborative approach where all parties win, which should also be achieved through a comprehensive examination of possible solutions. Therefore, in this paper, we aim to identify the most appropriate solution to the industrial development dispute in the Markazi province using the cooperative game theory model.

Unlike many conflict-resolution approaches that usually consider the interests of individual organizations and use noncooperative game theory methods such as (Graph Model for Conflict Resolution II), the innovation of this research is to solve the problem using cooperative methods, focusing on social choice rules (SCRs) and fallback bargaining (FB).

In fact, unlike noncooperative approaches, which only explain the situation of conflict between actors, the cooperative approach tries to identify ways out of conflicts by emphasizing the positions and demands of the stakeholders involved in the dispute—to indirectly reduce the effects and pressures of conflicts and act to improve organizational performance. Moreover, applying a cooperative approach to land use conflicts, especially industrial development, has not received much attention so far, an issue that in developing countries, while guaranteeing sustainable incomes for governments, also puts pressure on the environment.

In this regard, the most important question of this research that must be answered is what will be the most important approach to the cooperative model of game theory regarding the industrial development conflict in the Markazi province?

Moreover, based on studies, the following two hypotheses can be considered for this research. Hypothesis 1: Given the complexity of industrial development tensions in the Markazi province, it seems that government intervention to resolve this issue will be one of the main options on the table. Hypothesis 2: Moreover, considering the nature of the methods used in the collaborative game theory model in this research, the fallback bargaining (FB) approach will work better than the social choice rules (SCRs) approach to solve this conflict.

## 2. Materials and Methods

### 2.1. Study Area

The study area is the Markazi province (34.6123° N, 49.8547° E), which is located in the center of Iran and covers an area of about 29 127 km$^2$ with cold and dry climate. The Markazi province has 1.429 million inhabitants, of which 76.9% live in urban areas. The population density in the Markazi province is 49.1 people per square kilometer, which is almost the same as Iran [39]. Due to the special geopolitical location of the Markazi province in Iran and its proximity to the major cities of the country, this region is the industrial center of Iran and many important industries such as refineries, petrochemicals, aluminum, locomotive manufacturing, etc. are located in the Markazi province. On the other hand, due to these infrastructures, Arak city (center of the Markazi province) is

ranked the 71st most polluted city in the world [40] and is one of the most polluted cities in Iran.

In general, although the Markazi province is better off than many other provinces in Iran in terms of economic conditions and infrastructure, the lack of attention to the local population and the environment has led to an increase in social discontent. Thus, the migration trend in the province has reversed in recent years, and more people tend to leave the province rather than live in it. In fact, the problem of industrial growth and development and environmental pollution in the Markazi province has become a complex problem that requires the consideration of the interests of all stakeholders.

To investigate the usefulness of game theoretic models for collective decision making in land development processes, we developed a game model for a case study in which an industrial area was developed in Iran's Markazi province.

*2.2. Methods*

In general, the type of research method used in this research is analytical-descriptive. In this way, in addition to illustrating what is, the reasons for how it is and why the situation of the problem and its dimensions will be explained [41]. In this regard, to explain and justify the reasons, it needs a strong argument. This support is provided by searching the literature and theoretical topics for research and formulating general propositions and theorems about it. In fact, in this research, we will try to logically connect the details of the research problem with the relevant general propositions and draw conclusions. When the purpose of the research is to identify the problem, analyze and classify the variables (in this research actors and their goals), descriptive-analytical research is a good choice. Using this type of research is useful when there is not much information about the subject or issue of the research and no studies have been done so far [42].

To conduct this research, which took place in the winter of 2021, literature reviews, newspaper surveys, semi-structured interviews with experts from the organizations involved, and upstream documents were used to carefully interview all individuals and organizations involved in the conflict over industrial development in the Markazi province.

Unlike noncooperative game theory, which analyzes the actions and payoffs of individual players, cooperative game theory studies the collective actions and resulting collective payoffs of a group of players (or coalitions) [43]. A cooperative game is a mathematical structure formed by a group of players, a coalition, who can achieve a joint payoff that enforces cooperative behavior [44–46]. In terms of measuring mutual effects between actors, cooperative game theory is considered an effective and valid method for equitably distributing costs and benefits to ensure system stability and sustainability [47]. Two well-known subclasses of cooperative games are the social choice rule (SCRs) and fallback bargaining rule (FB) classes (Figure 2) [48,49].

The processes of this model include five main steps: designing the group of decision makers, identifying all possible conflict situations based on the previous step, eliminating impossible conflict situations, determining the preferences of the actors, and implementing the model to identify possible equilibria and resolve conflicts [50].

### 2.2.1. Players

In the end, the most influential actors include Department of Environment (DOE), Industries and Mines Organization (IMO), and Plan and Budget Organization (PBO). One of the main reasons for selecting these actors is the top-down political and administrative system in Iran, which prevents maximum consideration and involvement of the people, especially the local population, in important decisions.

In reality, there are more actors involved in the process of industrial development, e.g., the provincial government, interest groups such as nongovernmental organizations, small-scale industry and the organization of industrial towns, landowners and developers in other competing locations. However, we only consider these three in our game because they are the crucial actors in this process.

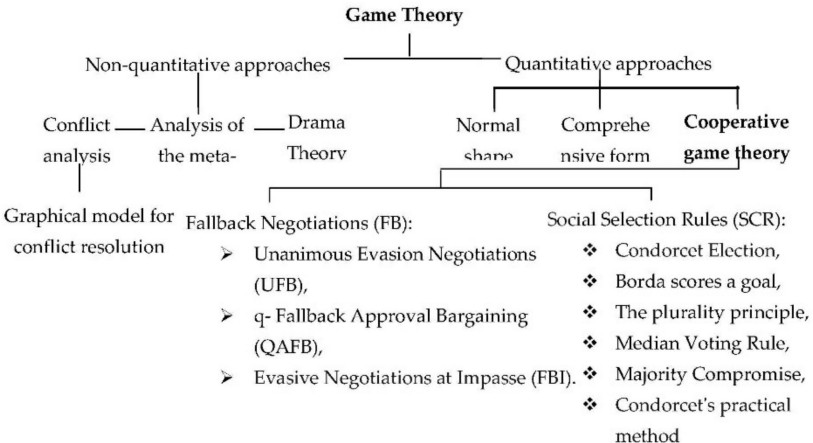

**Figure 2.** Classification of the different models of game theory [51] (with a slight modification).

### 2.2.2. Distributions

In the game model presented in this study, we estimated the payoffs at the interval level for each possible outcome of the game. To this end, we first conceptualized the goal of the industrial development process considered by IMO into three subgoals that are consistent with payoff maximization. These subgoals are (1) the financial outcome of the development process, (2) the equitable distribution of opportunities across all parts of the Markazi province, and (3) the time period in which the plan is implemented ("the sooner the better"). However, the importance of these three different benefits varies for different stakeholders. For DOE, "distribution of opportunities" and "time" are more important than "financial outcome", while for developers this is probably the other way around. The reason for this is that the DOE believes that there is no place for further industrial development due to the many environmental problems in the province. Therefore, it considers organizing the current situation more important than other options. The role of the PBO as a facilitator is to strategically plan and monitor the development of the Markazi province by conducting scientific studies to improve the planning and budgeting systems. The PBO has adopted a conservative approach to industrial development in the Markazi province. The reasons are firstly that the PBO aims at economic growth and development of the province. Secondly, because of the poor performance of the province's industrial targets and the problems they have caused so far, it wants to put more pressure on the environment and therefore on the people of the area.

These differences will be weighted differently for each player compared to these three subgoals. Each player will evaluate all outcomes against these three subgoals based on their preferences. For this evaluation, each player will use different variables according to their preferences and reflect the relevance of these variables in relation to each of the three sub-objectives of land and real estate development (Table 1).

### 2.2.3. Negotiation Modeling

In the negotiation and cooperation model, we assume that DOE and IMO agree on possible solutions by negotiating with each other rather than competing with each other and making independent decisions. Therefore, the methods of negotiation analysis, including Social Choice Rules (SCRs) and Fallback Bargaining (FB), are used to determine the most likely outcome of the negotiation. Because the bargaining process is not structured, the solution concept models the outcome of bargaining between actors and considers the value that each coalition can create [52].

**Table 1.** Stakeholders and variables in the conflict over industrial development in the Markazi province.

| Interested Parties | Variables |
|---|---|
| DOE | approval of industrial development in the event of compliance with the desired environmental reforms |
| | complete rejection of industrial development |
| | Petition the judiciary for temporary suspension of industrial projects pending DOE approval. |
| IMO | abolition of industrial development (organization of the current situation) |
| | industrial development taking into account environmental aspects |
| | industrial development without consideration of environmental aspects |
| PBO | financing of industrial development projects if the development is approved by DOE |
| | financing of industrial development projects without development permit by DOE |

The following options were considered in these negotiations:

Unbridled version of construction (UC): this option describes the situation where IMO industrial zones will be expanded without reducing their negative impacts despite DOE opposition to industrial development;

Compromise Coordination (C): by involving senior government officials as mediators in the conflict, the IMO commits to implementing environmental reforms to reduce the negative impacts of industrial development, and the DOE agrees to the IMO's construction of the industrial site;

Suspension through arbitration (Sa): in order to prevent the negative consequences of the construction of the industrial estate, high-ranking government officials intervene in the conflict and force the IMO to cancel the project, although they fully support the DOE's positions;

Mitigated version of construction through arbitration (TCa): after the IMO refused to implement the measures taken by the DOE to reduce the negative impacts of industrial construction, the legal system suspends the project until the DOE's corrections are received;

Suspension under environmentally friendly policy (Se): with the reform of the country's development planning system (in accordance with the Sixth National Development Plan of Iran), the issuance of licenses and the granting of loans for development projects by the PBO will be subject to the approval of the industrial development project by the DOE;

Mutual Compromise on Environmentally Friendly Measures (Ce): In this situation, the DOE agrees to industrial development in the Markazi province if the necessary changes to reduce negative impacts are complied with. The IMO also agrees to comply with the changes in order to obtain a permit and provide credit for the project.

Depending on the parties' position in the dispute and their objectives, they therefore have different priorities for the various options (Table 2).

**Table 2.** Preferences of parties regarding negotiation options for industrial land development in the Markazi province.

| Negotiating Parties | Preferences of the Parties with Regard to the Negotiation Options | | | | | | | | | | |
|---|---|---|---|---|---|---|---|---|---|---|---|
| DOE | Se | > | Sat | > | TCa | > | C | > | Ce | > | UC |
| IMO | UC | > | C | > | Ce | > | TCa | > | Sat | > | Se |

### 3. Results

#### 3.1. Social Choice Rules (SCRs) Method

Social Choice Rules (SCRs) studies the rules of choice in situations where, first, multiple decision makers have different and sometimes conflicting preferences regarding possible choices for themselves and, second, intend to make group decisions instead of individual decisions [39,48,53,54]. The purpose of the concepts in this theory is to combine the preferences of decision makers in such a way that they are reflected as equally and fairly as possible in the group preferences [55].

### 3.1.1. Condorcet Choice

The winner of the Condorcet process is a bargaining option that is at least as desirable to the majority of the bargaining parties as the other options [39]. In other words, the option that is least desirable to the majority of the negotiating parties is the optimal social mode based on the Condorcet choice rule [55]. To find the optimal Condorcet mode, all bargaining options should be considered pairwise (Table 3) for the bargaining parties. The option that receives the most benefits (preference or desirability) in the pairwise comparison is the winner or Condorcet optimum.

**Table 3.** Pairwise comparison of negotiation options in industrial land development in the Markazi province.

| Negotiation Options | UC | | C | | Sat | | Tca | | Se | | Ce | |
|---|---|---|---|---|---|---|---|---|---|---|---|---|
| | D | I | D | I | D | I | D | I | D | I | D | I |
| UC | - | - | L | W | L | W | L | W | L | W | L | W |
| C | W | L | - | - | L | W | L | W | L | W | W | W |
| Sat | W | L | W | L | - | - | W | L | L | W | W | L |
| TCa | W | L | W | L | L | W | - | - | L | W | W | L |
| Se | W | L | W | L | W | L | W | L | - | - | W | L |
| Ce | W | L | L | L | L | W | L | W | L | W | - | - |

Annotation. D: DOE, I: IMO, W: Won, L: Lost.

The results of Table 4 show that option C (compromise coordination) has the highest score of 6 points among the six negotiation options and is therefore the socially optimal negotiation for the industrial development of the Markazi province in terms of Condorcet's rule of choice.

**Table 4.** Calculate the points and determine the winner of Condorcet.

| | Negotiation Options | No. Victory | No. | Total Score (Total Winnings) |
|---|---|---|---|---|
| | UC | 5 | 5 | 5 |
| Condorcet winner | C | 6 | 4 | 6 |
| | Sat | 5 | 5 | 5 |
| | Tca | 5 | 5 | 5 |
| | Se | 5 | 5 | 5 |
| | Ce | 4 | 6 | 4 |

### 3.1.2. Borda in Goal

According to this scoring rule, in order to find a social optimum, the bargaining options must be scored based on the degree of desirability for each decision maker. By adding up the scores, the option with the highest score is called the Borda choice, or the social optimum in terms of the Borda rule. To find the Borda optimum, if (m) represents the total number of negotiation options, the option that is most desirable to a negotiating party is assigned the score (m-1). The second most preferred option is assigned the score (m-2) and the other options are scored in the same way. In other words, the score assigned to each option from the perspective of each decision maker under the Borda rule indicates how many other negotiating options are preferable to that negotiating party [55].

For example, the compromise coordination option (C) selected as the Borda optimum according to Table 5 has the fourth priority for DOE out of a total of six options and is preferred only compared to Ce and UC options.

**Table 5.** Calculation of points and determination of the optimal Borda option.

| | Negotiator<br>Negotiation Options | DOE | IMO | Total Score |
|---|---|---|---|---|
| **Borda<br>Optimal** | UC | 0 | 5 | 5 |
| | C | 2 | 4 | 6 |
| | Sat | 4 | 1 | 5 |
| | Tca | 3 | 2 | 5 |
| | Se | 5 | 0 | 5 |
| | Ce | 1 | 3 | 4 |

Therefore, in Table 5, option C is given a score of 2 (6 − 4 = 2). Moreover, for IMO, this option is the second most preferred. In other words, for IMO, option C has priority over the other four options and is only less preferred than option UC. Therefore, for IMO, this option is given a score of 4 (6 − 2 = 4). Because this option has a score of 6 and all other options have a score of 5 or less, Option C is optimal for Borda in the industrial development negotiations.

### 3.1.3. The Majority Rule

This rule is one of the simple and ancient rules used to determine the optimal social system. A plurality rule in voting means that one option has received more votes than the other, even if it has not reached an absolute majority or consensus. Therefore, this rule selects the option that is preferred by most of the negotiators [56]. Thus, it assigns a score of 1 to the option that is most preferred by a particular negotiator, and the decision maker gives a score of 0 to the other options. By doing this for all decision makers, the option that receives the most points is selected as the social optimum.

Therefore, it can be said that the selection method in this rule is such that it only cares about the option with the highest preference and disregards the lower priority options.

According to Table 6, the UC option is the first priority of IMO, and Se option is the first priority of DOE. Therefore, these two options receive a score of 1, and due to the fact that the other options have 0 points, both are selected as the optimal plurality of industrial development negotiations.

One of the weaknesses of the plurality rule is that the selected option may not be accepted by the majority of the negotiating parties; in other words, it may not have the maximum support of the parties [57].

**Table 6.** Calculation of points and determination of plurality optimal option.

| Negotiating Parties | | Preferences of the Parties with Regard to the Negotiation Options | | | | | | | | | |
|---|---|---|---|---|---|---|---|---|---|---|---|
| DOE | Plurality<br>Optimal | Se | > | Sat | > | TCa | > | C | > | Ce | > | UC |
| Majority points | | 1 | | 0 | | 0 | | 0 | | 0 | | 0 |
| IMO | Plurality<br>Optimal | UC | > | C | > | Ce | > | TCa | > | Sat | > | Se |
| Majority points | | 1 | | 0 | | 0 | | 0 | | 0 | | 0 |

This is especially true in cases where there is a great deal of conflict between the preferences of the negotiating parties over the options. This is true of the optimal plurality obtained for the industrial development negotiation. As can be seen, each of the solutions obtained is the best option of one party and the worst option of the other party. Therefore, the chances of the party's compromise on their choice as a solution are very low. Therefore, in such cases, based on the optimal plurality principal to avoid complicating the conflict and moving towards resolving it, two options are chosen as the acceptable option.



The following two rules are intended to solve this problem and select options that may not be the first priority of the negotiating parties but are supported by the majority of the negotiating parties and are more likely to be agreed upon.

### 3.1.4. Median Voting Rule

This rule selects as socially optimal the option that has the most negotiating votes (highest support) and the highest possible preference level [58]. Because the two parties (IMO and DOE) are involved in industrial development negotiations, the most supported option is the one with the highest preference level for both parties. The preference or support level of an option is determined by the position of that option in the preference order of each party to the negotiation.

According to Table 7, the compromise coordination (C) and moderate version of construction through arbitration (TCa) options of industrial development negotiations are chosen based on the median voting rule. These two options have been agreed upon by the parties at the fourth priority level, and although they are less preferred than the other more desirable options, they have majority support and are more likely to be agreed to.

**Table 7.** Calculation and determination of the optimal median choice option.

|         | Negotiation Options | Degree of Support from the Negotiating Parties | | | | | |
|---------|---------------------|---------|---------|---------|---------|---------|---------|
|         |                     | Stage 1 | Stage 2 | Level 3 | Level 4 | Level 5 | Level 6 |
|         | UC                  | 1       | 1       | 1       | 1       | 1       | 2       |
| Optimal | C                   | 0       | 1       | 1       | 2       | 2       | 2       |
|         | Sat                 | 0       | 1       | 1       | 1       | 2       | 2       |
| Optimal | Tca                 | 0       | 0       | 1       | 2       | 2       | 2       |
|         | Se                  | 1       | 1       | 1       | 1       | 1       | 2       |
|         | Ce                  | 0       | 0       | 1       | 1       | 2       | 2       |

### 3.1.5. Majority Compromise

This rule, like the median voting rule, selects as socially optimal an option that enjoys the support of the majority of the negotiating parties at the highest possible preference level. However, the only difference with the median voting rule is that the equality between multiple options that enjoy majority support is removed based on the stronger majority criterion [59].

In other words, among the options supported by the majority, the option supported by the largest number of negotiators is selected as the optimal option. Therefore, we can say that majority compromise is a form of the median voting rule, and the response obtained with this rule is a subset of the responses obtained with the moderation rule.

Moreover, the majoritarian compromise rule tries to find the Pareto optimum, an option that is acceptable to the majority of negotiators and better than other options. Therefore, this rule never chooses the option that is the worst candidate for the strict majority of negotiators [57].

As Table 7 shows, both "C" and "TCa" which are optimally selected by median voting rule have consensus in addition to majority support of negotiators. This is because there are two main sides to the negotiation of the conflict over industrial development in the Markazi province, and therefore majority support implies consensus among the negotiators. Therefore, both options are considered optimal in terms of majority compromise.

### 3.1.6. Condorcet's Practical Rule

In this rule, the option that is not supported by the majority of the negotiating parties at the first preference or support level is selected. If none of the options receives the maximum support at the first preference level, the option that has the highest number of supporters over the other options at the second preference level is selected [60].

Because this option is the most preferred option, in order to break the impasse, the lack of maximum support for the higher preference options is subtracted from the criterion

of needing absolute majority support for the high preference option and is satisfied with relative majority support.

As Table 2 shows, none of the industrial development negotiation options are supported by a majority of the parties (both parties) at the first preference level. However, at the second level, the four options C, UC, Sa and Se are supported by at least one of the parties and have a relative majority compared to the two options TCa and Ce. Therefore, options C, UC, Sa and Se are the optimal negotiated solutions based on the practical Condorcet rule.

### 3.2. Fallback Bargaining (FB) Method

Fallback bargaining (FB) [61] is a method for predicting the likely outcome of bargaining procedures. FB simulates the behavior of bargaining parties who fallback in lockstep from their most preferred solution or alternative to a less-desired solution until an agreement is reached.

### 3.2.1. Unanimous Evasion Negotiations (UFB)

Unanimity fallback bargaining (UFB) is a method for predicting the outcome of negotiations between two or more decision makers. In this method, the preferences of all parties for the negotiation options are first determined. Then, it is assumed that each negotiator starts with the option he or she prefers the most (maximum preference) and insists on it. The negotiating parties then gradually and progressively withdraw from their positions to reach an agreed solution [48].

This means that parties first retreat from the option that is their top priority to the option with the second highest priority and similarly to the options with lower preference. This process of retreat continues until they have the opportunity to gain the support of the majority or supermajority or, if possible, the unanimity of all negotiating parties [62].

Thus, the outcome of this method of negotiation indicates the point at which the greatest possible compromise is reached between the parties, which is why the outcome is called the compromise set.

When the condition or rule of consensus is used as a criterion for calculating the bargaining chip, the result is always Pareto-optimal. That is, the optimal option is the best option that can be accepted by all decision makers. However, such an optimal answer is not necessarily the preferred option of each decision maker, but most likely lies in the middle of the preferences of each negotiator. Therefore, it can be said that this negotiation method aims to minimize the desirability and satisfaction of each negotiator as much as possible [61].

As Table 8 shows, when seeking a compromise-based solution using unanimity fallback negotiation (UFB), the option with the highest preference for each party is considered first. If the option is the same for all parties (both DOE and IMO), the negotiation process is stopped and that option is recognized as the compromise solution. In this case, the depth of agreement is equal to one. The concept of depth of agreement refers to the level of preference and support of the conflicting parties at which a compromise is reached [63]. Thus, when the depth of agreement is equal to one, the option that is most preferred by all negotiating parties is chosen as the compromise solution. If no compromise is reached at the first preference level, we move to the second preference level and examine the second priorities of the negotiating parties.

**Table 8.** Calculation of the negotiation compromise using the unanimity solution (UFB).

| Negotiator | | | | | | | | |
|---|---|---|---|---|---|---|---|---|
| DOE | Se | ·······▶ | Sat | ·······▶ | TCa | ·······▶ | C | Ce | UC |
| Preferential rate | 1 | | 2 | | 3 | | 4 | 5 | 6 |
| IMO | UC | ·······▶ | C | ·······▶ | Ce | ·······▶ | TCa | Sat | Se |
| Preferential rate | 1 | | 2 | | 3 | | 4 | 5 | 6 |

*Depth of consent*

According to Table 8, DOE's fourth priority is option (C), which was also IMO's second priority. Therefore, option (C) is at least the fourth priority of the two parties in the industrial development conflict negotiations. On the other hand, option (TCa), which is IMO's fourth priority, is also DOE's third priority. Therefore, option (TCa) is at least the fourth priority of both parties in industrial development negotiation. Because two consensus compromise solutions were found in the industrial development negotiations in the depth of the fourth agreement (d = 4), the negotiation process was stopped, and the compromise options are "C" and "TCa". They are selected as a set of compromises in the industrial development negotiations using the UFB (Unanimity fallback bargaining) rule.

### 3.2.2. q-Approval Fallback Negotiations (QAFB)

The process Q consent fallback (QAFB) negotiation rule is very similar to the UFB rule. The difference is that in negotiations based on the QAFB rule, the condition for choosing an option as a compromise solution and the agreement of the negotiating parties is not consensus or a majority of votes, but reaching a certain number of votes determined by the analyst. Thus, the purpose of this rule is not to find a solution that is acceptable to all or a majority of the negotiating parties, but to identify a solution that is acceptable and acceptable to only a certain number of negotiating parties [61].

Applying the QAFB rule allows the analyst to increase the range of possible responses to the problem by reducing the degree of acceptance and agreement from consensus and majority to a particular minority or even one of the negotiating parties. However, the likelihood that such solutions will be accepted in the real-world changes as the degree of acceptance increases or decreases. For example, if there are five negotiators in a negotiation, q (the number of negotiators who accept the compromise solution) must be equal to five to reach a consensus solution. In this case, the solution found will be accepted by all negotiators and will most likely be implemented in the real world, and the answer obtained will be similar to the answer obtained from the UFB.

In the industrial development negotiation with two negotiating parties (the DOE and the IMO), when q = 2, the compromise solution must necessarily be accepted by both parties, at the highest possible preference level. In this case, the compromise solution consists of two options, (C) and (TCa), and the depth of agreement is equal to four, which is exactly the result of the UFB rule (Table 8). However, when q = 1, the option that has the highest preference for one of the parties (not both) is the compromise solution. In this case, the options (UC, IMO priority) and (Se, DOE priority) are the answer, and the depth of agreement is equal to one.

### 3.2.3. Deadlocked Catch-Up Negotiations (FBI)

As mentioned above, each negotiating party starts the negotiations with the option they prefer the most (maximum preference). The negotiating partners then gradually and step-by-step withdraw from their positions in order to find a mutually agreeable solution and identify the option that is most acceptable to all parties. However, a minimum

preference or "impasse" can be defined for each decision maker. The impasse is an option that the negotiator does not want to back down from, and if this is not the case, he prefers to leave the negotiation. To negotiate with the FBI (Fallback Bargaining with Impasse) rule, the option(s) preferred by each party are defined, and in the preference matrix, these options are distinguished from the lower preference options by marking I = Impasse. The compromise set is then calculated in the same way as previously described for the UFB rule [61].

In industrial development negotiations, the worst possible situation for DOE is industrial development without compliance with environmental reforms. Therefore, this decision maker may not be willing to accept an agreement that includes this option (UC). Therefore, the impasse for DOE is industrial development with environmental improvements. The worst situation for the IMO is if the construction project for the industrial development is canceled for some reason. Therefore, for IMO, they are unlikely to accept an agreement that includes the abandonment of the dam construction (options Sa and Se). Therefore, for IMO, the impasse is over the construction of an industrial estate with environmental reforms. Once the impasse of the parties is determined, the negotiation process begins. Table 9 shows how the compromise solution negotiated by industrial development can be calculated using the impasse rule.

**Table 9.** Calculation of the negotiation compromise using the FBI rule (Fallback Bargaining with Impasse).

| **Negotiator** | | | | | | | | |
|---|---|---|---|---|---|---|---|---|
| DOE | Se | Sat | TCa | C | | Ce | I | UC |
| Preferential rate | 1 | 2 | 3 | 4 | | 5 | (Dead end) | 6 |
| IMO | UC | C | Ce | TCa | I | Sat | | Se |
| Preferential rate | 1 | 2 | 3 | 4 | (Dead end) | 5 | | 6 |

(DOE: "Dead end" rotated label under column I. IMO: "Depth of consent" rotated label under column TCa; "Dead end" rotated label under column I.)

A look at Table 9 shows that if q (the number of negotiators who accept a compromise solution) is equal to one, the negotiation has no compromise solution. This is because the first priority of each party is lower than the impasse of the other party. However, if q = 2, we can look for a consensus solution.

If no agreement is reached by the third level of support, the parties are again forced to retreat to the fourth level of their preferences. DOE's fourth priority is option (C), which was also IMO's second priority. Therefore, option (C) is at least the fourth priority of the two industrial development negotiators. On the other hand, option (TCa), which is the fourth priority of IMO, is also the third priority of DOE. Thus, option (TCa) is at least the fourth priority of both negotiating parties for industrial development. After two consensus compromise solutions were found in the negotiations on industrial development in the depth of the fourth agreement (d = 4), the negotiation process was stopped and options (C) and (TCa) were selected.

### 3.3. Summarizing the Results

In this research, six common SCRs rules—Condorcet Choice, Borda Scoring, Plurality, Median Voting, Majority Compromise, and Condorcet's Practical—were used to identify

an option that disputants could agree upon if they were willing to make and negotiate group decisions.

Based on the results, options (UC) and (Se) were selected as socially optimal according to the two rules of plurality and Condorcet's practice. These two rules assume that obtaining a relative majority of the negotiating parties is sufficient to determine the agreed option. Note, however, that each of these two options is the first priority of one of the negotiating parties and the last priority of the other party. Thus, the Untempered Version of Construction (UC) option is the best preference of IMO and the worst preference of DOE. On the other hand, the Environmentally Friendly (Se) suspension option is also DOE's best preference and IMO's worst preference. Therefore, the likelihood of agreement between the two parties on either of these options seems very low.

The option of the moderate variant of construction through arbitration (TCa) was recognized as a possible agreed option by the two rules of median voting and majority compromise. The condition for choosing an option as optimal in terms of these two rules is to obtain an absolute majority of the votes of the negotiating parties, although the said optimality is not the first preference of the parties. Accordingly, the option (TCa) at the fourth preference level was supported by both negotiating parties (IMO and DOE). However, the results generally show that the compromise coordination option (C) is recognized as the social optimum of conflict negotiation in the Markazi province by five of the six rules used (Table 10). The only plurality rule that selects only the best preference of the negotiating parties as optimal does not select option (C) as the possible agreed option. Therefore, from SCRs' perspective, the most likely option to be agreed upon in the industrial development negotiations between DOE and IMO is compromise coordination (C).

**Table 10.** Results of bargaining analysis with Social Choice Rules (SCRs) and Fallback Bargaining (FB).

| | Social Selection Rules (SCRs) | | | | | | Fallback Negotiations (FB) | | | |
| | | | | | | | | QAFB | | |
| | Condorcet Election | Borda Hits | Plurality | Median Vote | Majority Compromise | Condorcet's Practical | UFB | Q = 1 | Q = 2 | FBI |
|---|---|---|---|---|---|---|---|---|---|---|
| UC | | | * | | | * | | * | | |
| C | * | * | | * | * | * | * | | * | * |
| Sat | | | | | | * | | | | |
| TCa | | | | * | * | | * | | * | * |
| Se | | | * | | | * | | * | | |
| Ce | | | | | | | | | | |

This option reflects the circumstances in which senior government officials, acting as mediators in the conflict, are advising IMO to implement the industrial development policy reforms demanded by DOE and, in turn, encouraging DOE to agree to the expansion of the industrial area if the environmental reforms are met.

As mentioned above, the basic concept of the fallback bargaining rule is to find the option with the highest probability of reaching a compromise between the negotiating parties. Such an option may not necessarily be the option preferred by all parties because it is based on moving away from maximum positions and agreeing on the option that is most desirable to the parties. This is particularly evident in the Markazi province industrial development negotiations because the maximum preferences (the most preferred options) of the DOE and IMO are in stark contrast because each party's first priority is lower than the acceptable preference of the other party, and if the parties insist on their first priority, no agreement is possible.

The results of the Fallback Bargaining (FB) rule in three different forms therefore show that the most appropriate options for reaching agreement in industrial development negotiations are "TCa" and "C".

Under either option, IMO will modify the industrial development plan based on DOE's proposed changes. Conversely, DOE, by deviating from its position of complete rejection

of the industrial development, agrees to the implementation of the project, provided the amendments are complied with. Such a compromise is mediated by senior government officials under Option "C" and by judicial mediation under Option "TCa".

Moreover, the results of the statistical analysis between the six bargaining options show that the standard deviation between these options is 2.62. Moreover, the standard deviation between the social selection rules (SCRs) is 1 and between the alternative bargaining rules (FB) is equal to 0.

## 4. Discussion

Game theory is one of the best-known methods for resolving conflicts, especially in the field of land development. This model predicts probabilities and equilibrium states by analyzing the behavior and decisions of the active and involved players [10]. Each game involves actors, strategies, and the benefits of a particular decision. In other words, game theory attempts to predict the behavior of actors based on their immediate goals in the conflict [64].

Cooperative game theory models how players compete and cooperate to create and preserve value in unstructured interactions as coalitions. The difference between cooperative and noncooperative models is that cooperative game theory focuses on how much players can acquire given the value that each coalition of players can create, while noncooperative game theory focuses on what moves players should rationally make. Note that "cooperative" and "noncooperative" are technical terms and do not represent an assessment of the degree of cooperation between players in the model [34].

In this paper, we use the cooperative game theory model and its main bargaining analysis methods, including social choice rules (SCRs) and fallback bargaining (FB). Social choice rules (SCRs) are mathematical and statistical concepts used for optimal social identification. These concepts anticipate the possible options on which they agree, assuming that the disputants intend to make group decisions rather than individually. Each of these concepts uses different rules to predict the outcome of the negotiation. However, the aim of all the concepts is to identify—among the different and sometimes conflicting preferences of the negotiating parties regarding the negotiation options—the option that has the most common ground for all parties.

So far, there are many strategic studies on land development [2,9,33,37,38,65–72] from urban to rural areas to changing the nature of land use and development policies, most of which use the noncooperative approach and also rarely focus on the conflicts in industrial development, especially through the cooperative model. One of the differences between this research and other researches is its scope, so this article studied and analyzed 10 rules on 6 options, focusing on the main actors (not all actors), and it was comprehensive.

Some studies have been focused on the applications of the game theory to address industrial development or land-use management problems [9,33,37]. Compared to such studies, clearly the results vary from study to study such that the figures are not of the same order of magnitude. The reason for this difference might lie in differences among the assumptions underlying the different condition and models employed.

Results indicated the success of the proposed method for industrial development tension programs that could be replicated for other disputes in such field. It was also shown that simultaneous application of the social choice rules (SCRs) and fallback bargaining (FB) models to industrial development management led to significantly reduced uncertainties while the acceptability of treatment strategies also increased significantly.

Moreover, this paper attempts to consider all solutions to conflict through negotiation and even withdrawal, rather than prioritizing organizational positions and interests. Monetary and financial instruments (through PBO) have also been used to encourage the parties to negotiate.

Reaching a compromise (albeit against the interests of the parties) instead of stopping plans and helping forces such as complaints was another notable achievement of this paper, unlike other studies. Perhaps the most important achievement of this study compared to

other similar studies is the consideration of sustainable development goals in this study, so that environmental issues were discussed along with economic problems and the results are ultimately beneficial to the local people and the environment.

In response to the main question of this research, it should be stated that option "C"—which includes the involving of senior government officials as mediators in the conflict, with the highest scores—has been introduced as the most important approach to resolve this conflict. Moreover, the first hypothesis is accepted because, as it was predicted, due to the complexity of the issue and the lack of solution in the province in recent years, the use of a stronger force that is somewhat convincing actors is necessary to consider that according to the results of this research, a senior government official actor (option "C") would be a better option compared to other options. Moreover, it is very difficult to reject or confirm the second hypothesis because, based on the results of this research, the score that each proposed option to solve this conflict has gained from the subsets of these two approaches (FB or SCRs) is equal and one of these two approaches cannot be preferred over the other.

On the other hand, as with many studies based on the cooperative approach of game theory, one of the weaknesses of this research is its quality, which leads to the researcher's opinion being incorporated into the study process. Another problem that overshadows this qualitative research is the fact that the studies are based on the most recent decisions made by the organizations involved, which may change completely if there are changes in the management system, ruling parties or upstream rules in Iran. In the past, decision makers did not pay special attention to environmental issues, but today, with the emergence of environmental issues in Iran, consideration of the environment is one of the most important parameters for decision making. It should also be noted that determining the type of rules for each option is based on studies, but stakeholder experts may not agree.

## 5. Conclusions

The inability of classical optimization methods to resolve environmental conflicts led to the use of multi-objective methods such as e-constraint, goal planning, and trade-off planning in the early 21st century, and later game-theoretic approaches replaced earlier approaches.

Due to the wide range of conflicts, a variety of methods of game theory have been developed so far, and on this basis, various classifications have been made in this field. Cooperation and negotiation methods, which are among the most commonly used models of game theory, are coherent methods for the analysis of real-world conflicts and can be used to study many different issues in the field of environmental policy, especially industrial land management.

Conflict negotiation for industrial development in the Markazi province follows a cooperative approach, i.e., it is assumed that instead of individual and competing decisions, the parties involved in the conflict try to make decisions as a group and in consultation with each other. For this purpose, the social choice rules (SCRs) and the fallback bargaining rule (FB) were used. The Social Choice Rules (SCRs) aim at integrating the different and sometimes conflicting preferences of the decision makers in such a way that, from the set of possible decision scenarios (negotiation options), a subset is identified that has more common ground for all parties. The Fallback Bargaining (FB) rule also follows the goal of finding the option that is most desirable for all negotiating parties. The result of these methods shows the most suitable options for compromise and agreement between the parties.

The results of this study show that the most likely option for agreement and compromise between the DOE and IMO on the industrial development dispute in the Markazi province is a government-brokered compromise (Compromise Coordination (C)). This is because it was selected as the answer to the problem by the largest number of mathematical concepts used.

The results shown can be used as a model for other conflicts in Iran and especially for other disputes related to common land resources. The findings can also be a guide

for industrial and environmental decision makers who should focus on negotiation and cooperation in a conflict rather than insisting on their firm and sometimes destructive position. In general, it seems that the key to resolving such complex disputes can be solved by two important policy-related tools: (1) open access to information and awareness of all actors involved about the goals and activities of related organizations; (2) policies and goals coherence at the macro level of the organization in order to achieve consensus and agreement and to satisfy all stakeholders involved.

It was also suggested that future research in this area should include a comparison between collaborative and noncooperative approaches, in addition to the inclusion of effective actors and leadership options. Moreover, because many such disputes are due to the lack of proper and integrated policy making at the macro levels, it is recommended to apply game theory at these levels in order to integrate policies visually.

One of the most important limitations of applying game theory and its models in such research is the modeling of actors, their interests and interactions with each other during different years. In fact, accurate modeling of many of these interactions is possible only through careful review of the upstream documents and reports provided; some of them are not necessarily available for reviewing and studying due to confidentiality.

To rectify this problem, creating a government-level document database in an impartial organization and making them freely available to all individuals can guarantee the resolution of such disputes through realistic modeling. Such a database can even significantly contribute to the organizational policies' coherence to prevent future tensions by providing relevant studies in the past.

Furthermore, the main limitation of this study is not considering all the actors involved in this conflict because identifying their priorities and the interactions between them would be very time consuming and would greatly increase the volume of modeling. Therefore, only the main actors should be satisfied.

Finally, conflicts are not bad per se, but to some extent they can provide an opportunity for negotiation and help identify aspects that need attention, growth and development. Apart from the outcome of the cooperative models of industrial development dispute, the resolution of the conflict in this case requires compromise between the parties to break the existing impasse. This can also be achieved in three ways: (a) the tendency to accept the credibility of the other party, even if they have a different point of view; (b) trying to understand the other perspective and vision, even if it contradicts our approach; (c) finding solutions that can meet the needs of all parties.

**Author Contributions:** Conceptualization, S.Z. and C.F.; methodology, A.H.A.; software, S.Z.; validation, S.Z., C.F. and A.H.A.; formal analysis, A.H.A.; investigation, S.Z.; resources, A.H.A.; writing—original draft preparation, A.H.A.; writing—review and editing, C.F.; visualization, S.Z.; supervision, C.F.; project administration, S.Z. All authors have read and agreed to the published version of the manuscript.

**Funding:** This research received no external funding.

**Institutional Review Board Statement:** Not applicable.

**Informed Consent Statement:** Not applicable.

**Conflicts of Interest:** The authors declare no conflict of interest.

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
