# Peer review of "Optimal Politics of Conflict over Physical-Industrial Development Using a Technique of Cooperative Game Theory in Iran"

_sustainability, doi:10.3390/su132212716_

Round 1

Reviewer 1 Report

With the increasingly fierce economic and social background, the topic of this manuscript is particularly interesting and in my view it must be analysed as soon as possible. It needs the following minor revisions:

1. Please explain the reasons why the manuscript analyze the optimal politics of conflict over physical-industrial development using a technique of cooperative game theory in Iran? I think this is extremely important.
2. The manuscript's analysis of previous research on this subject is inadequate. In my opinion, literature review should be an independent chapter.
3. Innovation is a key component of the paper and should be emphasized in the manuscript.

Author Response

Dear Reviewer

We would like to thank you for your helpful and constructive comments that contributed to the richness of this paper.

Reviewer 2 Report

Necessary corrections:

1. In the Abstract there is a lack of the aim of the study. In order to interest potential readers it is advisable to write applicative results of the study.

2. The introduction needs to be re-formulated. First, I suggest to divide it into smaller, thematic parts, so called subchapters.

3. The Introduction lacks many important quotations. For example, after the first sentence, which, by the way, is too long, it must be a literature reference (line: 37). The same comment applies to lines: 41, 45, 50, 52, 72, 75.

4. Such things like explanation of basics (line: 78), f. ex. "game theory" should go before explaining it, which means one paragraph earlier.

5. There is a lack of a fundamental, substantial part of the article, called: Materials and methods!

6. There are plenty of methods which can be used for the purpose of the paper.

7. The materials and methods are not sufficiently explained.

8. There is no information why those research methods have been chosen.

9. Where are the research questions?

10. Where is / are hypothesis?

11. Where is the confirmation/rejection of the hypothesis?

12. Under what criteria/s the variables have been selected (table 1, lines: 305-6-307)?

13. How the optimal plurality of industrial development negotiations can be more deeply explained? (line: 405).

14. The part called results needs an extensive re-formulation. It needs to be organized well into thematic subchapters, according to the researched item/s.

15. There should not be literature quotation in the Conclusion (lines: 680-685).

16. The Conclusion part needs to show results, sorted out in a different perspective, manner, goal, etc.

 17. What are the limitations of the methods used in the paper?

 18. How these limitations can be decreased/omittted?

 19. What are the limitations of the undergone research?

20. How, in a practical way, the results of the study can be used by Iranians (lines: 71-705)?

21. What do authors understand under the wording of “compromise between the parties” (line: 712)?

22. The literature needs to be enriched, especially by the findings from the theoretical sphere of the study.

Positives:

The article tackles with a very important and interesting subject.

Author Response

Dear Reviewer

We would like to thank you for your helpful and constructive comments that contributed to the richness of this paper.

Rebuttal Notes for Manuscript sustainability-1415730

Reviewer(s)' Comments to Author:

Reviewer: 2

Comments to the Author

  1. In the Abstract there is a lack of the aim of the study. In order to interest potential readers it is advisable to write applicative results of the study.

Agree. Please check the below explanation and text.

Goal: Therefore, the main purpose of this paper is to study the industrial development dispute in Markazi province through a cooperative model of game theory in order to conflict resolution through a comprehensive approach.

Additional explanation for result: In fact, the results of this study indicate that if the actors involved in this conflict do not cooperate to resolve it, it can lead to more complex problems and the involvement of other groups who may not even have a proper perception of the conflict.

  1. The introduction needs to be re-formulated. First, I suggest to divide it into smaller, thematic parts, so called subchapters.

Agree. According to the above constrictive comments we added two sections to the introduction: 1-2- Literature review; 1-3- The goal of study. Also to complete some gaps and this sections we added some explanations as well. Please check the text.

  1. The Introduction lacks many important quotations. For example, after the first sentence, which, by the way, is too long, it must be a literature reference (line: 37). The same comment applies to lines: 41, 45, 50, 52, 72, 75.
  2. Such things like explanation of basics (line: 78), f. ex. "game theory" should go before explaining it, which means one paragraph earlier.

Agree. First of all, we would like to thank you for your attention and help to improve the quality of the paper.

The points you have made in this regard have been addressed in the text, but we certainly accept it as one of the weaknesses of this research, because despite the extensive studies of the team of authors, unfortunately the number of studies conducted to cover all variables of this research (Industrial Development, Environment, Game Theory, Cooperative Model) are not so much. 

  1. There is a lack of a fundamental, substantial part of the article, called: Materials and methods!
  2. There are plenty of methods which can be used for the purpose of the paper.

We apologize and we may not understand you mean clearly. The Materials and Methods section is in the second part of this paper and at the end of the introduction section, which is divided into two parts: the study area and the method.

  1. The materials and methods are not sufficiently explained.
  2. There is no information why those research methods have been chosen.

Agree. We add the below explanation to the methods section of the paper.

In general, the type of research method used in this research is analytical-descriptive. In this way, in addition to illustrating what is, the reasons for how it is and why the situation of the problem and its dimensions will be explained. In this regard, to explain and justify the reasons, it needs a strong argument. This support is provided by searching the literature and theoretical topics for research and formulating general propositions and theorems about it. In fact, in this research, we will try to logically connect the details of the research problem with the relevant general propositions and draw conclusions. When the purpose of the research is to identify the problem, analyze and classify the variables (in this research actors and their goals), descriptive-analytical research is a good choice. Using this type of research is useful when there is not much information about the subject or issue of the research and no studies have been done so far.

To conduct this research, which took place in the winter of 2021, literature reviews, newspaper surveys, semi-organized interviews with experts from the organizations involved, and upstream documents were used to carefully interview all individuals and organizations involved in the conflict over industrial development in Markazi province.

We use of the two below references to support and complete the method section:

Fontenay, G,D. 2008. Analytical method transfer: New descriptive approach for acceptance criteria definition. Journal of Pharmaceutical and Biomedical Analysis, Volume 46, 1(7), 104-112.

Lepenioti, K., Bousdekis, A., Apostolou, D., Mentzas, G. 2020. Prescriptive analytics: Literature review and research challenges. International Journal of Information Management. 50, 57-70.

  1. Where are the research questions?

Agree. You right, since the below question added to the end of introduction section.

In this regard, the most important question of this research that must be answered is what will be the most important approach to the cooperative model of game theory regarding the industrial development conflict in Markazi province?

Response to the question (discussion section):

In response to the main question of this research, it should be stated that option “C”, which include of involving senior government officials as mediators in the conflict, with the highest scores, it has been introduced as the most important approach to resolve this conflict.

  1. Where is / are hypothesis?

We agree. The below explanation added to the end of introduction section in form of Hypothesis of this research.

Also, based on studies, the following two hypotheses can be considered for this research. Hypothesis 1: Given the complexity of industrial development tensions in Markazi province, it seems that government intervention to resolve this issue will be one of the main options on the table. Hypothesis 2: Also, considering the nature of the methods used in the collaborative game theory model in this research, the fallback bargaining (FB) approach will work better than the social choice rules (SCR) approach to solve this conflict.

  1. Where is the confirmation/rejection of the hypothesis?

Response to the hypothesis (discussion section):

The first hypothesis is accepted, because as it was predicted, due to the complexity of the issue and the lack of solution in the province in recent years, the use of a stronger force that is somewhat convincing actors is necessary to consider that according to the results of this research, a senior government officials actor (option “C”) would be better option compared to other option. Also, it is very difficult to reject or confirm the second hypothesis, because based on the results of this research and the score that each proposed option to solve this conflict has gained from the subsets of these two approaches (FB or SCR) is equal and one of these two approaches cannot be preferred over the other. To know.

  1. Under what criteria/s the variables have been selected (table 1, lines: 305-6-307)?

Agree. According to your previous comments, we added explanation to the “Method” section and especially the below explanation clarify your comment:

To conduct this research, which took place in the winter of 2021, literature reviews, newspaper surveys, semi-strutted interviews with experts from the organizations involved, and upstream documents were used to carefully interview all individuals and organizations involved in the conflict over industrial development in Markazi province.

It is worth to mention that, respectfully as you better than us know, in the Game theory studies to shape the model, Identifying actors, their interests, and prioritizing them should be determined through library studies and review of documents and resources.

  1. How the optimal plurality of industrial development negotiations can be more deeply explained? (line: 405).

Agree. Kindly attention to this explanation.

This is especially true in cases where there is a great deal of conflict between the preferences of the negotiating parties over the options. This is true of the optimal plurality obtained for the industrial development negotiation. Because as can be seen, each of the solution obtained is the best option of one party and the worst option of the other party. Therefore, the chances of the party’s compromise on their choice as a solution are very low. Therefore, in such cases, based on the optimal plurality principal to avoid complicating the conflict and moving towards resolving it, two options are chosen as the acceptable option.

  1. The part called results needs an extensive re-formulation. It needs to be organized well into thematic subchapters, according to the researched item/s.

Agree. According to your comment we divided the result section in to the 3 main section:

3.1. Social Choice Rules (SCR) Model

3.2. Fallback Bargaining (FB) Method

3.3. Summarizing the Results

  1. There should not be literature quotation in the Conclusion (lines: 680-685).

Agree. Many thanks, corrected.

  1. The Conclusion part needs to show results, sorted out in a different perspective, manner, goal, etc.
  2. What are the limitations of the methods used in the paper?

Agree. We add below explanation to the conclusion section. Kindly check.

One of the most important limitations of applying game theory and its models in such research is the modeling of actors, their interests and interactions with each other during different years. In fact, accurate modeling of many of these interactions is possible only through careful review of the upstream documents and reports provided, that some of them are not necessarily available for reviewing and studying due to confidentiality.

  1. How these limitations can be decreased/omittted?

Agree. We add below explanation to the conclusion section. Kindly check.

To rectify this problem, creating a government-level document database in an impartial organization and making them freely available to all individuals can guarantee the resolution of such disputes through realistic modeling. Such a database can even significantly contribute to the organizational policies coherence to prevent future tensions by providing relevant studies in the past.

  1. What are the limitations of the undergone research?

Agree. We added below explanation to the conclusion section. Kindly check.

As stated in the “2.2.1. Players” section of this study, the main limitation of this study is not considering all the actors involved in this conflict, because identifying their priorities and the interactions between them will be very time consuming and will greatly increase the volume of modeling. Therefore, only the main actors should be satisfied.

  1. How, in a practical way, the results of the study can be used by Iranians (lines: 71-705)?

Agree, please check the below explanation.

In general, it seems that the key to resolving such complex disputes can be solved by two important policy-related tools: 1) open access to information and awareness of all actors involved about the goals and activities of related organizations; 2) policies and goals coherence at the macro level of the organization in order to achieve consensus and agreement and to satisfy all stakeholders involved.

  1. What do authors understand under the wording of “compromise between the parties” (line: 712)?

Generally; “To compromise is to make a deal between different parties where each party gives up part of their demand. In arguments, compromise is a concept of finding agreement through communication, through a mutual acceptance of terms—often involving variations from an original goal or desires.”

Since, compromise here means cooperating to get out of this exit, even at the cost of losing a little bit of interest and reforming future processes.

  1. The literature needs to be enriched, especially by the findings from the theoretical sphere of the study.

Agree. According to your valuable opinion in the introduction section, a literature review has also been added. Kindly check the text.

From Authors: “Finally, we would like to thank you for your helpful and constructive comments that contributed to the richness of this paper."

Reviewer 3 Report

Thank you for the opportunity to revise the paper titled: “Optimal politics of conflict over physical-industrial development using a technique of cooperative game theory in Iran”

First of all, I would like to say that this paper addresses an interesting topic and the paper is well written. I do have, however, some concerns and suggestions for the authors:

The introduction does a good job at discussing what the paper is about, but the research question is not clearly stated. Please include a clear research question and describe how this question contributes to the literature by filling a gap and how this gap is important to be addressed.

The results are interesting, but they are not too much linked to previous literature. Only in a couple of places between discussion and conclusions it is referred to previous studies. It would be convenient to describe how your paper is linked to previous literature, how it advances specific fields and theories, to what extent it expands, contradicts, or moderates previous findings, etc.

Please avoid the repetition of the expression “The results of this study…” as it is used too many times in the conclusions.

Also, can you elaborate on the limitations of your research, as well as the future avenues for further research that arise from your results (at the moment there is only a very small mention to future studies in the conclusions, but much more could be developed).

Please check the indent of paragraphs.

Good luck!

Author Response

Dear Reviewer

We would like to thank you for your helpful and constructive comments that contributed to the richness of this paper.

Rebuttal Notes for Manuscript sustainability-1415730

Reviewer(s)' Comments to Author:

Reviewer: 3

Comments to the Author

  1. In the Abstract there is a lack of the aim of the study. In order to interest potential readers it is advisable to write applicative results of the study.

Agree. Please check the below explanation which added to the abstract section.

Therefore, the main purpose of this paper is to study the industrial development dispute in Markazi province through a cooperative model of game theory in order to conflict resolution through a comprehensive approach.

In fact, the results indicate that if the actors involved in this conflict do not cooperate to resolve it, it can lead to more complex problems and the involvement of other groups who may not even have a proper perception of the conflict.

  1. The introduction does a good job at discussing what the paper is about, but the research question is not clearly stated. Please include a clear research question and describe how this question contributes to the literature by filling a gap and how this gap is important to be addressed.

 Agree. Many thanks for your valuable comment. Since to enhance the introduction section based on you and another reviewer comment we re-arrange this section and added question and even hypothesis to the end of introduction.

In this regard, the most important question of this research that must be answered is what will be the most important approach to the cooperative model of game theory regarding the industrial development conflict in Markazi province?

Also, based on studies, the following two hypotheses can be considered for this research. Hypothesis 1: Given the complexity of industrial development tensions in Markazi province, it seems that government intervention to resolve this issue will be one of the main options on the table. Hypothesis 2: Also, considering the nature of the methods used in the collaborative game theory model in this research, the fallback bargaining (FB) approach will work better than the social choice rules (SCR) approach to solve this conflict.

Moreover, the below explanation are links to the responses of question and hypothesis in the “discussion” section.

In response to the main question of this research, it should be stated that option “C”, which include of involving senior government officials as mediators in the conflict, with the highest scores, it has been introduced as the most important approach to resolve this conflict. Moreover, The first hypothesis is accepted, because as it was predicted, due to the complexity of the issue and the lack of solution in the province in recent years, the use of a stronger force that is somewhat convincing actors is necessary to consider that according to the results of this research, a senior government officials actor (option “C”) would be better option compared to other option. Also, it is very difficult to reject or confirm the second hypothesis, because based on the results of this research and the score that each proposed option to solve this conflict has gained from the subsets of these two approaches (FB or SCR) is equal and one of these two approaches cannot be preferred over the other. To know.

  1. The results are interesting, but they are not too much linked to previous literature. Only in a couple of places between discussion and conclusions it is referred to previous studies. It would be convenient to describe how your paper is linked to previous literature, how it advances specific fields and theories, to what extent it expands, contradicts, or moderates previous findings, etc.

We agree with you, and unfortunately due to the lack of studies on the application of game theory in macro-industrial decisions and even industrial land use change, this issue can be considered as a weakness of this paper. But to cover this weakness and improve the quality of the paper, we have added a literature review section to the introduction and the related section in the results section has been strengthened.

 1-2- Literature Review

Various approaches have been applied in spatial land-use planning to model the interactions between land-uses and to reach a near-optimal decision. In recent research, optimization algorithms social choice rules (SCR) and fallback bargaining (FB) models of game theory have been further addressed. In this field, the most commonly used approach is that of employing cooperative algorithms [9, 33, 37, 38]. For example, In [9] demonstrates that game theory could help us to identify the key strategic decisions of land and property development projects by showing the different payoffs for stakeholders of their chosen strategies and selecting the equilibrium in which all stakeholders involved are best of; in [33], develops a new analytic framework from a behavioral perspective based on game theory. The framework concerns the logic and strategy of conflicts of legal land acquisition, and that of illegal land conversion; in [Error! Reference source not found.], seek to explore the underpinning logic of land-readjustment using fundamental concepts in cooperative game theory: the Shapley value and the Core. The results shed light on a range of important practical issues for the policy ranging from the conditions under which development might be self-initiated to coalition stability, and to the value of an animating agency. In those works, the game theory model was used in the optimization process in order to address some of the computational complexity involved, but such tools are not appropriate to deal with real industrial conflicts, because many game theory tools were incorporated into the industrial development goals analysis [29].

The main characteristic of the cooperative theory is the generation of various possibilities. This theory and their methods are usually adopted to optimize land-use layouts in a general way, without incorporating competitiveness. Based on the complexity of industrial development goals conflicts, the most common method for modeling through the cooperative game theory is social choice rules (SCR) and fallback bargaining (FB) [39]. In some studies, this tool is also utilized for land-use planning [9, 19, 23].

In addition, the below explanation added to the result section.

Some studies have been focused on the applications of the game theory to address industrial development or land use management problems [9, 33, 37]. Compared to such studies, clearly the results vary from study to study such that the figures are not of the same order of magnitude. The reason for this difference might lie in differences among the assumptions underlying the different condition and models employed.

Results indicated the success of the proposed method for industrial development tension programs that could be replicated for other disputes in such field. It was also shown that simultaneous application of the social choice rules (SCR) and fallback bargaining (FB) models to industrial development management led to significantly reduced uncertainties while the acceptability of treatment strategies also increased significantly.

  1. Please avoid the repetition of the expression “The results of this study…” as it is used too many times in the conclusions.

Agree. Many thanks, they are corrected.

  1. Also, can you elaborate on the limitations of your research, as well as the future avenues for further research that arise from your results (at the moment there is only a very small mention to future studies in the conclusions, but much more could be developed).

Agree. For further studies:

Also, since many such disputes are due to the lack of proper and integrated policy-making at the macro levels, it is recommended to apply game theory at these levels in order to integrate policies visually.

For limitations:

One of the most important limitations of applying game theory and its models in such research is the modeling of actors, their interests and interactions with each other during different years. In fact, accurate modeling of many of these interactions is possible only through careful review of the upstream documents and reports provided, that some of them are not necessarily available for reviewing and studying due to confidentiality.

To rectify this problem, creating a government-level document database in an impartial organization and making them freely available to all individuals can guarantee the resolution of such disputes through realistic modeling. Such a database can even significantly contribute to the organizational policies coherence to prevent future tensions by providing relevant studies in the past.

Furthermore, the main limitation of this study is not considering all the actors involved in this conflict, because identifying their priorities and the interactions between them will be very time consuming and will greatly increase the volume of modeling. Therefore, only the main actors should be satisfied.

  1. Please check the indent of paragraphs.

Yes, sure.

From Authors: “Finally, we would like to thank you for your helpful and constructive comments that contributed to the richness of this paper."

Round 2

Reviewer 2 Report

Dear Authors,

it is a pleasure to read the corrected version of your manuscript.

I have no further comments on your paper.

Best wishes and good luck !

Author Response

Authors revised according to all reviewers' comments.

Reviewer 3 Report

No further comments

Author Response

(The authors gave the same response as above.)
